# THINK PROPRIOCEPTIVELY: COMPACT SUBGOAL TRACES FOR VISION-LANGUAGE-ACTION MODEL

## ABSTRACT

Vision-language-action (VLA) models translate visual observations and language instructions to robot actions, yet current architectures regard proprioception as a passive input rather than an active reasoning component. Without proprioceptive guidance, VLA models process multimodal features in isolation from the robot's physical configuration, and hierarchical approaches often encode subgoals in high-dimensional visual or textual spaces that are ungrounded in the robot's embodiment. We present SubgoalVLA, a framework built on the *think proprioceptively* paradigm that redefines how multimodal information is processed. SubgoalVLA leverages proprioception in two ways. First, proprioceptive states serve as cross-attention queries to select vision-language features, enabling configuration-aware feature extraction. Second, subgoals are encoded as compact sequences of joint configurations that eliminate the need for cross-modal translation. Through a two-stage training protocol that begins with supervised learning on ground-truth subgoals and then fine-tunes with self-predicted subgoals, we mitigate distribution shift between training and inference. On the CALVIN benchmark, SubgoalVLA achieves state-of-the-art performance with an average task completion length of 3.32, demonstrating that proprioceptive reasoning provides the critical bridge between high-level task understanding and embodied control.

## 1 INTRODUCTION

Effective robotic manipulation requires the seamless integration of high-level semantic understanding with precise motor control, yet this challenge persists despite advances in learning-based methods. Vision-language-action (VLA) models (Zitkovich et al., 2023; Kim et al., 2025; Black et al., 2024) have emerged as a promising direction, translating visual observations and natural language instructions into robot behaviors through large-scale pretraining. These models, however, neglect the internal sense of joint configuration and movement dynamics. In the absence of proprioceptive guidance, these models process vision-language representations independently of the robot's physical configuration, overlooking critical connections between what is perceived, what is instructed, and what can be executed from the current state. And hierarchical VLA models that incorporate subgoals typically rely on high-dimensional visual or textual representations, which are not grounded in the robot's embodiment and requires cross-modal translation. This raises a question: can placing proprioception at the core of VLA reasoning, what we call *think proprioceptively*, enable both interpretable hierarchical planning and improved task performance?

Current VLAs treat the proprioceptive state as a passive input rather than leveraging it to guide multimodal processing. Single-system models, illustrated in Figure 1 (a), encode joint configurations as text tokens alongside visual and linguistic inputs (Zhao et al., 2025a; Kim et al., 2025), thereby treating all modalities uniformly within a language model. This design creates a fundamental mismatch between continuous control signals and discrete symbolic tokens. Dual-system models, illustrated in Figure 1 (b), (c), and (d), attempt to couple vision–language backbones with specialised action heads through mechanisms such as special tokens, feature pooling, or interleaved attention (Li et al., 2024a; Black et al., 2024; Shukor et al., 2025). Yet these models still process multimodal features in a uniform manner, without filtering them according to what the robot can manipulate from its current configuration. Using the robot's kinematic state as an active query offers a principled alternative, enabling configuration-aware feature selection.

Figure 1: Comparison of VLA architectures: (a) single-system, (b) special-token bridged, (c) feature-aggregation bridged, (d) interleaved-attention bridged, and (e) proposed SubgoalVLA.

Beyond feature selection, hierarchical VLA models face persistent challenges in representing sub-goals. Recent approaches decompose complex tasks through either textual descriptions or visual goal images, following *think textually* (Wen et al., 2024) or *think visually* (Zhao et al., 2025a) paradigms. Textual subgoals are interpretable to humans but lack the spatial precision and temporal dynamics required for reliable control. Visual subgoals capture richer spatial structure but necessitate expensive generation of high-dimensional images and reliance on specialised image-synthesis modules. Both strategies operate in representational spaces disconnected from the robot's embodiment, thereby requiring costly translation to generate executable actions.

To leverage proprioception for both feature selection and subgoal representation, we introduce Sub-goalVLA, a hierarchical framework that implements the *think proprioceptively* paradigm and re-thinking how VLA models process and reason over multimodal information. First, kinematic states are served as cross-attention queries to actively select features from visual–language representations, enabling configuration-aware information extraction. Second, subgoals are represented as compact joint sequences, termed subgoal traces, which capture both target configurations and the motion dynamics. This design is inspired by biological motor planning, in which humans imagine movements kinesthetically rather than static visualization.

We adopt a two-stage training paradigm that first applies supervised learning with ground-truth sub-goals and then fine-tunes on self-predicted subgoals to mitigate distribution shift between training and inference. On the CALVIN benchmark, SubgoalVLA attains a 44.7% success on 5-step task chains and an average completion length of 3.32, outperforming both subgoal-based methods and end-to-end VLA baselines. These results demonstrate that proprioceptive reasoning, coupled with robust hierarchical training, offers a scalable pathway toward interpretable robotic manipulation.

## 2 RELATED WORKS

### 2.1 VISION-LANGUAGE-ACTION MODELS

The main challenge in VLA models lies in bridging the semantic richness of language with the precision required for continuous action. Two major paradigms have developed to address this gap, each adopting a different treatment of the proprioceptive state.

**Single-system VLAs** integrate perception, reasoning, and action prediction within a single back-bone. Models such as RT-2 (Zitkovich et al., 2023) and OpenVLA (Kim et al., 2025) employ action tokenisers to discretise continuous motor commands (Zhao et al., 2025b;a), enabling direct super-vision of robot actions with language models. These architectures encode proprioceptive states as text tokens by converting joint angles and end-effector poses into symbolic representations. Al-though this unified design benefits from end-to-end optimisation, it introduces a mismatch between continuous control and discrete token representations.

**Dual-system VLAs** separate high-level semantic reasoning from low-level control by bridging mechanisms. Three principal bridging strategies have emerged. Special-token models, such as CogACT (Li et al., 2024a) and MoLeVLA (Zhang et al., 2025), compress multimodal features into reserved embeddings, which inevitably lose fine-grained details. Feature-aggregation models, in-

cluding DeeR (Yue et al., 2024) and ChatVLA (Zhou et al., 2025), apply pooling operations to select features while disregarding task-specific context. Interleaved cross-attention models, such as $\pi_0$(Black et al., 2024), GR00T-N1 (NVIDIA et al., 2025), and SmolVLA (Shukor et al., 2025), directly attend action policies to multimodal features, which enables flexible integration but incurs quadratic computational cost. The proprioceptive state is treated as a passive conditioning input rather than as an active mechanism for selecting features most relevant to the robot's configuration.

## 2.2 SUBGOAL-ORIENTED ROBOTIC MANIPULATION

Subgoal-based planning decomposes complex manipulation tasks into tractable segments, enabling more robust execution and interpretable failure recovery than end-to-end approaches (Chane-Sane et al., 2021; Andrychowicz et al., 2017). This principle has evolved from early waypoint extraction toward learned policies that generate and execute subgoals (Wang et al., 2025a).

**Foundation model-based subgoal planning.** Recent work has leveraged foundation models to strengthen subgoal reasoning. Diffusion based methods, such as SuSIE (Black et al., 2023a) and SkillDiffuser (Liang et al., 2024), synthesise visual subgoals that capture informative intermediate states. Language-based approaches (Wang et al., 2025b) instead generate textual subgoals that provide high-level semantic guidance. Although these methods demonstrate strong reasoning capability, their subgoals are represented in external modalities, either visual or textual, that remain disconnected from the robot's proprioception space.

**Subgoal-oriented VLA models.** The incorporation of subgoal reasoning into VLA models marks the latest advance in hierarchical robotic control. Approaches such as DiffusionVLA (Wen et al., 2025), CoT-VLA (Zhao et al., 2025a), and $\pi_{0.5}$ (Black et al., 2025) exemplify this trend by conditioning action generation on textual or visual subgoals. While the *think textually* and *think visually* paradigms underscore the importance of explicit intermediate reasoning, their high-dimensional representations impose substantial computational costs, reliance on specialised processing modules limits architectural flexibility, and the translation from visual or textual spaces to executable actions introduces additional error.

## 3 PROBLEM FORMULATION

Given a robotic manipulation task specified by a natural language instruction $l$, the goal is to enable a robot to infer interpretable subgoal traces and to generate actions that satisfy the instruction. We adopt behavioral cloning (Torabi et al., 2018) to learn directly from demonstration data without explicit reward engineering.

**MDP formulation.** We formulate the problem as a Markov Decision Process with tuple $(\mathcal{S}, \mathcal{A}, \mathcal{T}, l, \rho_0)$. At time $t$, the state $s_t \in \mathcal{S}$ contains visual and proprioceptive inputs. The visual observation $o_t = [I_t^1, \ldots, I_t^n]$ consists of RGB images from $n$ cameras. The proprioceptive state $q_t$ encompasses the robot's full kinematic configuration, including end-effector position and orientation, gripper width and action state, and joint angles. Actions $a_t \in \mathcal{A}$ specify relative end-effector poses with translation $\Delta T$ and rotation $\Delta R$. The transition function $\mathcal{T} : \mathcal{S} \times \mathcal{A} \to \mathcal{S}$ governs environment dynamics, and $\rho_0$ denotes the initial state distribution.

**Temporal representations.** To enable smooth control and hierarchical planning, we adopt chunked representations for both actions and subgoals. Following recent VLA work (Mees et al., 2024; Huang et al., 2025; Kim et al., 2025), an action chunk $A_t = (a_t, \ldots, a_{t+N-1})$ groups $N$ consecutive motor commands. For hierarchical planning, we introduce subgoals $g_t \in \mathcal{G}$ that represent intermediate targets toward task completion. Whereas existing approaches typically encode subgoals as images or textual descriptions, we instead define subgoal traces $G_t = (g_t, \ldots, g_{t+M-1})$ that capture motion dynamics

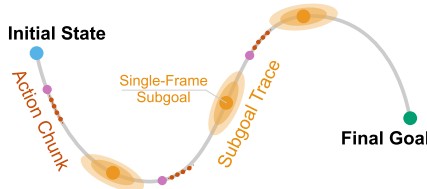

Figure 2: Subgoal traces and action chunks along a trajectory.

over $M$ steps. This temporal extension reflects how humans plan manipulation through kinesthetic simulation rather than static snapshots. In our framework, we instantiate subgoals in the robot's proprioceptive space, making them compact, and directly grounded in the robot's embodiment.

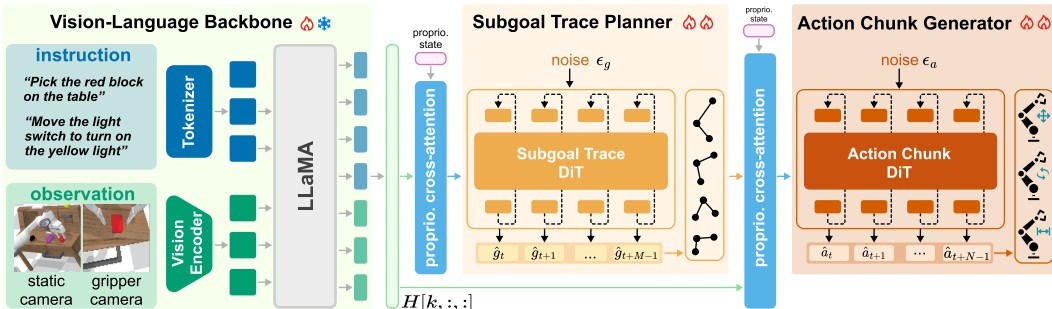

Figure 3: **SubgoalVLA overview.** The vision-language backbone encodes the current RGB observation and instruction into a joint representation. A DiT-based planner predicts an $M$-step proprioceptive subgoal trace that summarizes intended motion. A DiT-based generator denoises the next $N$ action commands conditioned on vision-language features and predicted subgoal trace.

**Learning objective.** Given demonstrations $\mathcal{D}$, each trajectory $\tau \in \mathcal{D}$ contains $(l, s_{1:T}, a_{1:T}, g_{1:T})$ with instruction, states, actions, and subgoals. We learn a hierarchical VLA policy $\pi_{\theta,\psi,\phi} : l \times \mathcal{S} \to \mathcal{A}$ through two-level control. The planner $\psi$ predicts subgoal traces, and the generator $\phi$ produces action chunks conditioned on these subgoals, both utilizing vision-language features from backbone $\theta$. Parameters are optimized through the imitation objective:

$$\theta^*, \psi^*, \phi^* = \arg\min_{\theta,\psi,\phi} \mathbb{E}_{\tau \sim \mathcal{D}} \left[ \mathcal{L}_g(\hat{G}_{1:T}; s_{1:T}, l) + \mathcal{L}_a(A_{1:T}; s_{1:T}, \tilde{G}_{1:T}, l) \right], \tag{1}$$

where $\mathcal{L}_g$ and $\mathcal{L}_a$ are the training objectives for the subgoal planner and action generator, respectively, and $\tilde{G}_{1:T}$ represents either ground-truth subgoals or predicted subgoals $\hat{G}_{1:T}$ depending on the training phase. Section 4.3 details this two-stage training strategy and its motivation.

# 4 METHODOLOGY

SubgoalVLA is a hierarchical policy that decomposes complex manipulation tasks through explicit proprioceptive reasoning. As illustrated in Figure 3, the framework integrates three specialized components that connect high-level task semantics with low-level execution. First, the **proprioceptive cross-attention mechanism** (§4.1) uses the robot's kinematic state to query multimodal embeddings, extracting features conditioned on the robot's configuration. Second, the **subgoal trace planner** (§4.2.1) generates compact proprioceptive traces that encode intended motion over a short horizon, providing interpretable intermediate structure. Third, the **action chunk generator** (§4.2.2) refines these plans into smooth executable control sequences, ensuring stability to robot dynamics. Finally, a **two-stage training strategy** (§4.3) is employed to train the entire model.

## 4.1 VISION-LANGUAGE BACKBONE WITH PROPROCEPTIVE CROSS-ATTENTION

We build on LLaVA (Liu et al., 2023), which combines CLIP (Radford et al., 2021) vision encoding with Vicuna (Chiang et al., 2023) language modeling. At time $t$, RGB observations $o_t$ are mapped to spatial features, flattened, projected through a learned adapter, and converted into visual tokens $\mathcal{V} = \{v_1, \ldots, v_P\}$ that are compatible with the language embedding space. The instruction $l$ is tokenized into textual tokens $\mathcal{T} = \{u_1, \ldots, u_L\}$. Visual and textual tokens are interleaved following LLaVA protocol and processed by the Vicuna transformer across $m$ layers. At each layer $k \in [1, m]$, the transformer produces contextualized hidden states $\mathbf{H}[k, :, :] \in \mathbb{R}^{(P+L) \times d}$, where $P + L$ is the combined sequence length and $d$ is the hidden dimension.

**Proprioceptive cross-attention.** To enable configuration-aware feature extraction, we introduce proprioceptive cross-attention that queries vision-language representations using the robot's kinematic state. Following GR00T-N1 (NVIDIA et al., 2025), we extract features from an intermediate layer (we use $k = m/2$ in experiments) to balance semantic abstraction with spatial grounding. The layer-$k$ hidden states $\mathbf{H}[k, :, :]$ provide keys $\mathbf{K}_{\text{VL}}$ and values $\mathbf{V}_{\text{VL}}$ for cross-attention.

The proprioceptive information is projected into the vision-language embedding space to form queries $\mathbf{Q}_{\text{proprio}}$. For the planner, these queries derive solely from the state history, maintaining kinematic consistency. For the generator, the queries additionally incorporate the predicted subgoal trace, allowing feature selection to reflect both current state and intended trajectory. Figure 4 illustrates the mechanism, with dashed lines denoting the trace specific queries used by the generator. These proprioceptive queries attend to the vision-language representations through multi-head cross-attention:

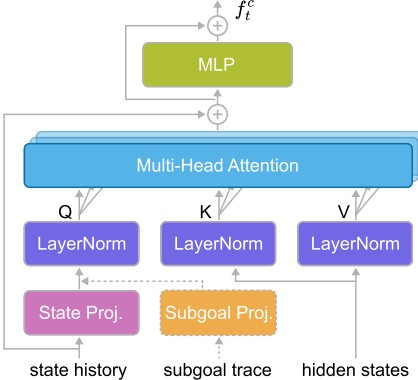

$$
\begin{aligned}
f_t^c &= \text{MHA}(\mathbf{Q}_{\text{proprio}}, \mathbf{K}_{\text{VL}}, \mathbf{V}_{\text{VL}}) + \mathbf{Q}_{\text{proprio}} \\
f_t^c &\leftarrow \text{MLP}(f_t^c) + f_t^c
\end{aligned} \tag{2}
$$

We follow standard transformer practice with prenormalization and dropout, which are omitted here for clarity. The dual residual connections stabilize fusion, one after cross-attention to preserve proprioceptive structure and one after the MLP to refine integrated features.

Figure 4: Proprioceptive cross-attention mechanism. The subgoal planner queries only proprioceptive state history, whereas the action generator also incorporates subgoal-trace queries, shown as dashed lines.

### 4.2 COMPLEMENTARY PLANNING AND EXECUTION

Bridging semantic reasoning with motor control requires models capable of both decomposing tasks and synthesising executable commands. SubgoalVLA addresses this challenge through two complementary Diffusion Transformer (DiT) modules that operate in distinct spaces. The planner functions in proprioceptive space to generate interpretable waypoints, while the generator operates in action space to produce smooth motor commands.

#### 4.2.1 SUBGOAL TRACE PLANNER

The planner generates subgoal traces $G_t$ over a horizon of length $M$, with each step encodes the robot's target proprioceptive configuration. These traces compactly represent motion intent, act as kinesthetic sketches of the planned trajectory.

We adopt conditional denoising diffusion for training. The forward process perturbs a clean trace $G_t$ as $G_t^{(i)} = \sqrt{\bar{\alpha}_i}\, G_t + \sqrt{1 - \bar{\alpha}_i}\, \epsilon_g$, where $\epsilon_g \sim \mathcal{N}(0, I)$, $i$ indexes the diffusion step, and $\bar{\alpha}_i$ denotes the cumulative noise schedule. Let $S_t \in \mathbb{R}^{h \times p}$ denote the proprioceptive state history over $h$ steps, where $p$ is the proprioceptive dimension encoding joint angles, end-effector pose, and gripper state. Proprioceptive queries $\mathbf{Q}_{\text{proprio}}^{\text{g}} = \text{Linear}(S_t)$ are derived solely from the proprioceptive state history. Cross-attention produces conditioning features $f_t^{\text{c,g}}$, which guide the DiT planner $\psi$ to predict noise as $\hat{\epsilon}_g^i = \psi(G_t^{(i)}, f_t^{\text{c,g}}, i)$. Training minimizes the prediction error between estimated and true noise:

$$
\mathcal{L}_g = \mathbb{E}_{\epsilon_g, i} \left\| \hat{\epsilon}_g^i - \epsilon_g \right\|_2. \tag{3}
$$

At inference, the planner is initialised with Gaussian noise and iteratively denoised to produce the final trace $\hat{G}_t$. This process yields subgoal predictions that are temporally coherent and grounded in the robot's proprioceptive dynamics.

#### 4.2.2 ACTION CHUNK GENERATOR

The generator converts subgoal traces into action chunks $A_t$ over horizon $N$, where each step denotes the relative end-effector pose. Whereas the planner captures high-level intentions, the generator ensures stable low-level execution.

Proprioceptive queries incorporate both current state and the subgoal trace: $\mathbf{Q}_{\text{proprio}}^{\text{a}} = \text{Linear}([S_t, \tilde{G}_t])$, where $[\cdot, \cdot]$ denotes concatenation and $\tilde{G}_t$ represents either ground-truth sub-

goals $G_t$ (Stage I) or predicted subgoals $\hat{G}_t$ (Stage II). Cross-attention yields conditioning features $f_t^{c,a}$ for action generation. During diffusion step $j$, the generator perturbs an action chunks as $A_t^{(j)} = \sqrt{\bar{\alpha}_j}\, A_t + \sqrt{1 - \bar{\alpha}_j}\, \epsilon_a$ with $\epsilon_a \sim \mathcal{N}(0, I)$. The DiT generator $\phi$ predicts noise according to $\hat{\epsilon}_a^j = \phi(A_t^{(j)}, f_t^{c,a}, j)$. The optimisation objective is defined as the prediction error between the estimated and true noise:

$$\mathcal{L}_a = \mathbb{E}_{\epsilon_a, j} \left\| \hat{\epsilon}_a^j - \epsilon_a \right\|_2. \tag{4}$$

At inference the generator starts from Gaussian noise and iteratively denoises to produce the final action sequence $\hat{A}_t$.

### 4.3 Two-Stage Training Paradigm

SubgoalVLA is trained in two phases that progressively align planning with execution. The planner and generator always minimize the diffusion objectives $\mathcal{L}_g$ (Eq. 3) and $\mathcal{L}_a$ (Eq. 4) using Eq. 1, but differ in how the generator is conditioned on subgoals.

**Stage I: learning with ground-truth subgoals.** In the first stage, ground-truth subgoal traces $G_t$ serve as training targets for the planner and conditioning inputs for the generator. The generator's queries are $\mathbf{Q}_{\text{proprio}}^a = \text{Linear}([S_t, G_t])$. This allows the planner to acquire accurate proprioceptive predictions while the generator learns subgoal-to-action mappings under ideal conditions. All parameters $\{\theta, \psi, \phi\}$ are optimized jointly, with the pretrained backbone $\theta$ updated conservatively.

**Stage II: adaptation to predicted subgoals.** The second stage addresses the distribution shift that arises during inference by replacing ground truth subgoals with planner predictions $\hat{G}_t$. The generator is thus trained with conditioning features $f_t^{c,a}(\hat{G}_t)$, which is computed by using Eq. 2 with $\mathbf{Q}_{\text{proprio}}^a = \text{Linear}([S_t, \hat{G}_t])$. The backbone $\theta$ is frozen for stability while both DiT modules continue optimization. This two-stage design ensures that the generator can operate under imperfect subgoals while the planner is encouraged to produce actionable traces.

## 5 Experimental Results

To comprehensively assess SubgoalVLA, we conduct experiments to address four key research questions: **(1) Reasoning.** Can the model decompose high-level instructions into coherent proprioceptive subgoals? **(2) Representation.** Do subgoal traces provide advantages over single-step or visual alternatives? **(3) Feature selection.** How effective is proprioceptive cross-attention compared to pooling or unguided attention? **(4) Performance.** How does SubgoalVLA compare with state-of-the-art approaches in long-horizon manipulation?

### 5.1 Experimental Setup

**Benchmark and evaluation protocol.** We evaluate on the CALVIN Long-Horizon Multi-Task Language Control (LH-MTLC) benchmark (Mees et al., 2022b), in which agents must execute sequences of five manipulation subtasks described by natural language. Following prior work (Mees et al., 2022a; Brohan et al., 2022; Li et al., 2024b; Yue et al., 2024; Zhao et al., 2025b), performance is measured as the average successful sequence length (ranging from 0 to 5) across 1,000 task chains, which provides a robust assessment of long-horizon reasoning and control. Further details of the dataset are given in Appendix A.

**Model configuration.** SubgoalVLA employs LLaVA-1.5 as the vision–language backbone, combining CLIP ViT-L/14 features with a 7B LLaMA-2 language model. To preserve broad visual–semantic priors, the vision encoder is frozen during training. Both the subgoal planner and the action generator are implemented as diffusion transformers, each comprising 12 layers, 8 attention heads, and a hidden width of 768. Training uses the AdamW optimiser with an initial learning rate of $1 \times 10^{-4}$, decayed to $3 \times 10^{-6}$ by a cosine schedule. Details of the network structure are provided in Appendix B, and the hyperparameter settings are listed in Appendix C.

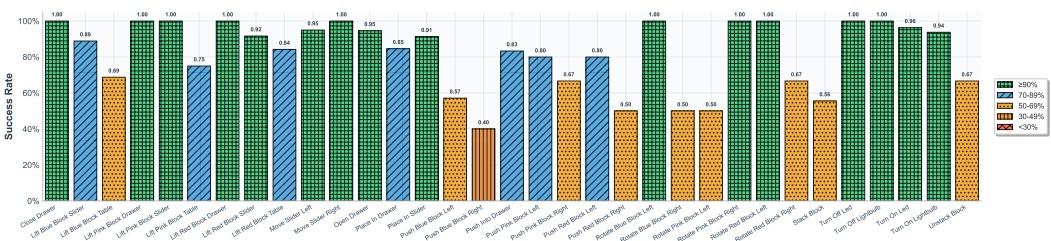

Figure 5: Task-specific success rates on CALVIN benchmark. SubgoalVLA performance across 34 manipulation tasks.

## 5.2 SUBGOAL-CONDITIONED ROBOT MANIPULATION

We now present experimental results on the CALVIN benchmark that analyze both the training dynamics and the task-level performance of SubgoalVLA.

**Task success rates.** Figure 5 presents per-task success rates across 34 CALVIN tasks, highlighting distinct performance trends. SubgoalVLA achieves near-perfect execution on slider movements ($> 95\%$) and drawer operations ($> 94\%$), demonstrating that proprioceptive subgoals effectively capture these well-defined kinematic trajectories. The model shows moderate performance ($50 - 70\%$) on block lifting and placement tasks, where outcomes depend jointly on reaching precision and grasp stability. Complex manipulation tasks such as stacking ($55.6\%$) and rotation tasks with specific blocks remain challenging, suggesting that certain task-object combinations require more nuanced proprioceptive planning.

**Two-stage training dynamics.** Figure 6 illustrates the optimization behavior across the two stages. In Stage I, both the subgoal planner and the action generator converge rapidly under ground-truth conditioning: the planner loss decreases from $\sim 1.0$ to below $0.05$ within the first epoch, while the generator loss decreases from $\sim 0.8$ to below $0.04$. This steep decline reflects the efficiency of joint training with shared proprioceptive cross-attention and vision–language features. In Stage II, the generator is conditioned on predicted subgoals to address distribution shift. The planner maintains a stable loss of $0.009 \pm 0.01$ with minimal fluctuation, while the generator continues to improve with a consistent reduction in loss. This asymmetric dynamic indicates that the planner produces robust subgoals and the generator adapts

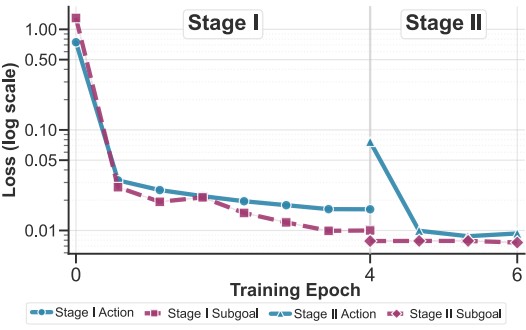

Figure 6: **Training dynamics of the two-stage paradigm.** Stage I (left) conditions on ground-truth subgoals, and Stage II (right) conditions on predicted subgoals.

effectively, showing that the two-stage paradigm mitigates exposure bias while preserving training stability.

**Subgoal visualization.** Figure 7 visualizes predicted proprioceptive subgoals rendered by replaying joint configurations directly in the simulator for representative manipulation tasks. For clarity, we show single frames sampled from the full subgoal traces, each capturing a critical waypoint in the task execution. The planner generates semantically meaningful intermediate states, such as aligning the end-effector with a drawer handle before pulling, or hovering above a block prior to grasping. These visualizations demonstrate that proprioceptive subgoals provide interpretable intermediate structure that directly corresponds to task-critical waypoints.

## 5.3 COMPARISON

We compare SubgoalVLA against a range of methods previously evaluated on the CALVIN benchmark. These include approaches that employ latent subgoals, such as the Hierarchical Universal

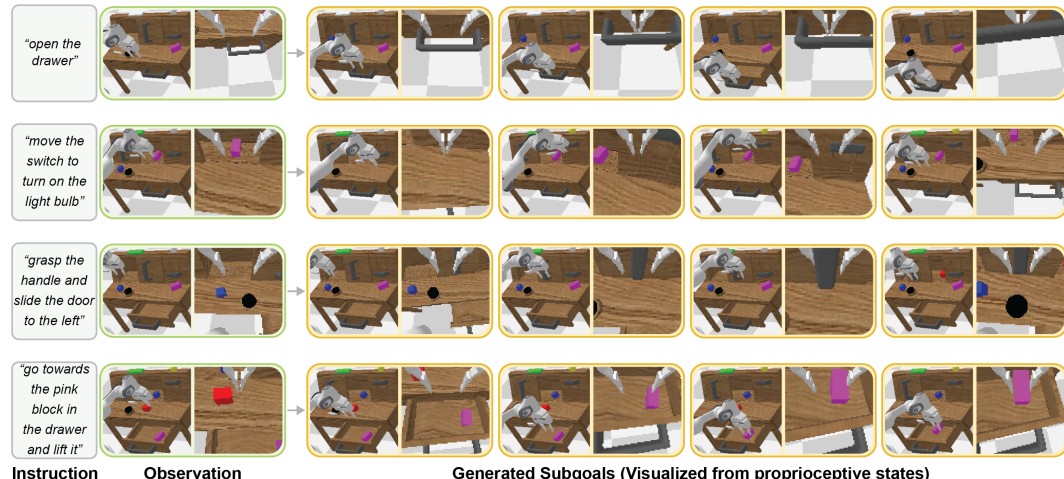

Figure 7: **Proprioceptive subgoal decomposition.** Each row shows the task instruction (*left*), the initial observation (*center*), and predicted proprioceptive subgoals rendered from static and gripper views (*right*).

Table 1: Long-horizon success rates (↑) on CALVIN (ABC → D). LH-$k$: task chain of length $k$; Avg. Len.: average successful chain length.

| Method | Category | Foundation Model | LH-1 | LH-2 | LH-3 | LH-4 | LH-5 | Avg. Len. |
|---|---|---|---|---|---|---|---|---|
| HULC (RA-L'22) | Latent | - | 0.418 | 0.165 | 0.057 | 0.019 | 0.011 | 0.67 |
| SuSIE (NeurIPS'23) | Image | InstructPix2Pix | 0.870 | 0.690 | 0.490 | 0.380 | 0.260 | 2.69 |
| RT-1 (RSS'23) | Single | - | 0.533 | 0.222 | 0.094 | 0.038 | 0.013 | 0.90 |
| VLAS+ (ICLR'25) | system | LLaMA-2 7B | 0.859 | 0.592 | 0.385 | 0.259 | 0.176 | 2.27 |
| RoboFlamingo (ICLR'24) | Dual | Flamingo-3B | 0.824 | 0.619 | 0.466 | 0.331 | 0.235 | 2.47 |
| DeeR (NeurIPS'24) | system | Flamingo-3B | 0.862 | 0.701 | 0.518 | 0.415 | 0.304 | 2.82 |
| **SubgoalVLA** | Dual system | LLaMA-2 7B | **0.875** | **0.771** | **0.667** | **0.563** | **0.447** | **3.32** |

Language-Conditioned policy (HULC) (Mees et al., 2022a), which learns hierarchical representations through feature-level adaptation. We further evaluate SuSIE (Black et al., 2023b), which generates image-based subgoals by leveraging a pretrained image-editing model to synthesize future visual states from language commands. In addition, we benchmark against recent state-of-the-art VLA models, comprising action-tokenizer-based single-system architectures such as RT-1 (Brohan et al., 2022) and VLAS+ (Zhao et al., 2025b), as well as dual-system architectures with specialized action heads, including RoboFlamingo (Li et al., 2024b) and Dynamic early-exit for Robotic models (DeeR) (Yue et al., 2024). All methods are trained and evaluated under identical visual and textual input conditions; for VLAS+, we adopt the text-based configuration rather than the speech-based variant to ensure fair comparison. We provide detailed baseline descriptions and additional experiments in Appendix D.2 and Appendix E.

Table 1 highlights systematic differences across paradigms. Latent planning methods such as HULC struggle beyond single-step tasks, with performance collapsing to 0.011 on LH-5. Image-based subgoals, exemplified by SuSIE, sustain longer sequences with a score of 0.260 on LH-5. Single-system vision–language–action models, including RT-1 and VLAS+, achieve strong short-horizon performance with success rates reaching 0.859, yet their performance degrades sharply as the horizon lengthens. Dual-system models, represented by RoboFlamingo and DeeR, further improve stability; DeeR attains 0.304 on LH-5. In contrast, SubgoalVLA achieves 0.875 on LH-1, surpassing SuSIE

at 0.870 and DeeR at 0.862, while maintaining superior results across horizons, reaching 0.447 on LH-5 and achieving the highest average sequence length of 3.32. These gains demonstrate that compact proprioceptive subgoals, directly aligned with the robot's control space, provide a scalable and interpretable foundation for long-horizon manipulation.

Table 2 compares subgoal representation dimensions across hierarchical methods. HULC employs 288-dimensional latent codes (256 for plan encoding and 32 for goal specification), whereas SuSIE generates full 256×256 pixel visual subgoals, amounting to 65,536 dimensions. By contrast, SubgoalVLA requires only 120 dimensions (15×8), capturing 15 temporal waypoints of 8-dimensional proprioceptive states. This represents a 546-fold reduction relative to visual approaches and a 2.4-fold reduction relative to latent methods, demonstrates the efficiency of proprioceptive representations, while retaining interpretability as direct joint configurations rather than abstract embeddings.

Table 2: Comparison of subgoal representation dimensions.

| Method | Subgoal Dim. |
|---|---|
| HULC | $256 + 32$ |
| SuSIE | $256 \times 256$ |
| SubgoalVLA | $15 \times 8$ |

### 5.4 Ablation Study

We conduct ablation experiments to evaluate the role of the subgoal planner in hierarchical reasoning and the mechanism for aggregating vision–language features.

**Subgoal planner.** We first assess the role of the subgoal planner by ablating either the entire module (**w/o SP**) or collapsing the multi-step *subgoal trace* into a single waypoint (**w/o ST**). As shown in Table 3, removing the planner forces the generator to rely directly on raw multimodal embeddings, reducing LH-5 success to 24.0%. Replacing traces with single subgoals also weakens performance (32.3% at LH-5). These results demonstrate that both the planner and temporally coherent traces are essential for sustaining multi-step reasoning, as a single static waypoint is insufficient to guide extended behaviors.

Table 3: Ablation of subgoal planning components.

| Method | LH-1 | LH-2 | LH-3 | LH-4 | LH-5 | Avg. Len. |
|---|---|---|---|---|---|---|
| w/o SP | 0.855 | 0.645 | 0.467 | 0.339 | 0.240 | 2.55 |
| w/o ST | 0.864 | 0.719 | 0.552 | 0.416 | 0.323 | 2.875 |
| **w ST** | **0.875** | **0.771** | **0.667** | **0.563** | **0.447** | **3.32** |

**Feature aggregation strategy.** We compare proprioceptive cross-attention with alternative aggregation schemes for token-level representations from the vision–language backbone. As shown in Table 4, static pooling leads to severe degradation: mean pooling almost collapses performance on LH-5 (0.0%), max pooling provides only marginal gains, and generic attention pooling remains far below usable levels. These results indicate that effective feature selection must be conditioned on the robot's proprioceptive state, which enables task-relevant tokens to dominate the multimodal representation.

Table 4: Ablation of feature aggregation strategies.

| Method | LH-1 | LH-2 | LH-3 | LH-4 | LH-5 | Avg. Len. |
|---|---|---|---|---|---|---|
| mean | 0.039 | 0.001 | 0.000 | 0.000 | 0.000 | 0.040 |
| max | 0.085 | 0.005 | 0.000 | 0.000 | 0.000 | 0.090 |
| attn. | 0.119 | 0.008 | 0.000 | 0.000 | 0.000 | 0.127 |
| **proprio. cross attn.** | **0.875** | **0.771** | **0.667** | **0.563** | **0.447** | **3.32** |

## 6 Conclusion

We have presented **SubgoalVLA**, a hierarchical framework that embodies the *think proprioceptively* paradigm for robotic manipulation. By employing proprioception both to query vision–language features through cross-attention and to represent subgoals as compact joint traces, SubgoalVLA enables interpretable hierarchical planning without the computational burden of visual or textual intermediates. A two-stage training strategy further facilitates robust deployment by mitigating the gap between training and inference. A current limitation is that the framework relies on ground-truth proprioceptive demonstrations for training, which constrains applicability in settings where only visual demonstrations are available.

ETHICS STATEMENT

This work uses only the publicly available CALVIN benchmark in simulation, involving no human subjects, private data, or animal studies. We are not aware of conflicts of interest or foreseeable negative societal impacts arising from this research.

REPRODUCIBILITY STATEMENT

We provide comprehensive details to ensure reproducibility. Dataset specifications (Appendix A) detail the CALVIN benchmark configuration, subgoal generation parameters, and preprocessing. Network architecture (Appendix B) specifies all components including CLIP ViT-L/14, LLaMA-7B, DiT-B modules, and proprioceptive cross-attention with exact configurations. Training hyperparameters (Appendix C) cover learning rates, batch sizes, optimizer settings, and two-stage training schedules. Implementation details include diffusion parameters, dimension specifications, and computational requirements. Code is available at the anonymous repository `https://anonymous.4open.science/r/wyrngg/` for review. Upon acceptance, we will release the complete codebase and pre-trained models to facilitate reproduction and future research.

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

## LLM USAGE

Large Language Models (LLMs) were used as a general-purpose writing and editing assistant during the preparation of this manuscript. Specifically, LLMs were employed to polish grammar, improve clarity, and suggest stylistic refinements to align with standard scientific writing conventions. The LLM did not contribute novel research ideas, experimental results, or substantive content beyond language editing and presentation support.

## A  DATASET

**CALVIN dataset.** The CALVIN dataset comprises four environmental splits (A–D), each defined by distinct backgrounds and object configurations. Each split contains more than 2 million robot manipulation trajectories (denoted 'ALL'), of which approximately 1%—about 24,000 trajectories—are annotated with language instructions (denoted 'LANG'). In this study, we adopt the ABC → D training paradigm, where models are trained on three environments and evaluated on the held-out target environment. Visual inputs from both static and wrist-mounted cameras are concatenated and resized to $334 \times 334$ pixels to enhance training efficiency.

**Subgoal generation.** Subgoal traces are defined by two parameters: the subgoal start frame and the subgoal trace duration. At each time step, the subgoal is constructed from the robot's proprioceptive

states, beginning at the designated start frame and extending for the specified duration. For trajectories in which the start frame exceeds the task horizon, we pad the trace by replicating the terminal state. As the average task length in CALVIN is approximately 60 time steps, we empirically set the subgoal start to 14 and the trace duration to 8.

# B  NETWORK ARCHITECTURE

SubgoalVLA adopts a hierarchical architecture that integrates a multimodal large language model backbone with proprioceptive cross-attention (PCA) modules and diffusion-based components for subgoal planning and action generation. The system comprises seven core components: (1) a CLIP ViT-L/14 image encoder that processes visual inputs from both static and wrist-mounted cameras, (2) a multi-layer perceptron (MLP) projector that maps visual features into the language model embedding space, (3) a LLaMA-7B language model that interprets task instructions and contextual information, (4) a proprioceptive cross-attention module (multi-head attention, MHA) for task-aware feature extraction, (5) a Diffusion Transformer (DiT) subgoal planner that predicts future proprioceptive state sequences, (6) a second proprioceptive cross-attention module for subgoal-conditioned action grounding, and (7) a DiT action generator that produces robot actions conditioned on current observations and planned subgoals. This modular design supports end-to-end learning while preserving interpretability through explicit subgoal representations.

| Component | Model | Hidden dim | # Layers | # Heads | Params |
|---|---|---|---|---|---|
| Image Encoder | CLIP ViT-L/14 | 1024 | 24 | 16 | 304M |
| Projector | MLP 2×GELU | 1024→4096 | 2 | - | 8M |
| LLM | LLaMA-7B | 4096 | 32 | 32 | 7B |
| PCA (VL $\rightarrow f_t^{c,g}$) | MHA | 4096 | 3 | 8 | - |
| Subgoal Planner | DiT-B | 768 | 12 | 12 | 86M |
| PCA (VL+$G \rightarrow f_t^{c,a}$) | MHA | 4096 | 3 | 8 | - |
| Action Generator | DiT-B | 768 | 12 | 12 | 86M |

Table 5: Specifications of SubgoalVLA components, including backbone models, cross-attention modules, hidden dimensions, and parameter sizes.

## B.1  SUBGOAL PLANNER

We instantiate a Diffusion Transformer (DiT) to predict subgoal traces conditioned on fused vision–language embeddings and proprioceptive state histories. Table 6 summarizes the hyperparameters and architectural specifications of the Diffusion Transformer subgoal planner. The planner generates proprioceptive subgoal traces of length 8, conditioned on a state history of 15 steps. The diffusion process employs 8 denoising steps with a cosine noise schedule parameterized by squared–cosine cumulative $\bar{\alpha}$, balancing training stability with sample diversity. The DiT backbone embeds subgoal tokens, timestep features (via an MLP applied to sinusoidal embeddings), and conditioning inputs, with label dropout ($p = 0.1$) to enable classifier-free guidance. Latent variable $z$ is concatenated before the subgoal tokens, and sinusoidal positional embeddings with a learned global scale are applied. Each block employs pre-LayerNorm self-attention and MLP layers modulated through AdaLN gates, computed from the timestep embedding combined with pooled $z$. The model is trained with timestep-weighted MSE between predicted and true noise (weighted by $1 - t/T$), together with a small $\ell_2$ regularization term ($\lambda = 0.001$) on predictions. Multiple timesteps are sampled per batch to improve efficiency during training. At inference, we adopt DDIM sampling and apply classifier-free guidance by mixing conditional and unconditional predictions with a guidance scale $s > 1$.

## B.2  ACTION GENERATOR

Table 7 shows the hyperparameters and training configuration of the action generator. The generator predicts action chunks of length 15, conditioned on both proprioceptive state histories (15

| Parameter | Value |
|---|---|
| Subgoal trace size | 8 |
| Diffusion steps | 8 |
| State history size | 15 |
| Noise schedule | Cosine schedule with squared cosine alpha bar |
| Loss function | MSE |
| Positional encoding | Sinusoidal |

Table 6: Hyperparameter configuration of the Diffusion Transformer subgoal planner.

steps) and planner-predicted subgoal traces (8 steps). Training follows an 8-step denoising diffusion process with a cosine noise schedule based on squared–cosine cumulative $\bar{\alpha}$. Sinusoidal positional encodings are applied to preserve temporal structure. The DiT backbone and conditioning layers (positional/AdaLN) are identical to those of the subgoal trace planner, ensuring architectural consistency across modules. The optimization objective is a mean squared error (MSE) loss between predicted and ground-truth noise, using repeated diffusion sampling. At inference time, we employ DDIM sampling with 10 steps and optional classifier-free guidance (CFG) to balance diversity and determinism. The DiT backbone and positional/AdaLN conditioning are identical to the subgoal trace planner.

| Parameter | Value |
|---|---|
| Action chunk size | 15 |
| Diffusion steps | 8 |
| State history size | 15 |
| Subgoal trace size | 8 |
| Noise schedule | Cosine schedule with squared cosine alpha bar |
| Loss function | MSE |
| Positional encoding | Sinusoidal |

Table 7: Hyperparameter configuration of the Diffusion Transformer action generator.

## C    HYPERPARAMETERS SETTING

**Hyperparameters.** SubgoalVLA employs a two-stage training paradigm with carefully tuned hyperparameters to ensure stable convergence and optimal performance across all model components. The training process utilizes differential learning rates for different architectural components, reflecting their varying complexity and pre-training status. The vision projector and language model employ lower learning rates (1e-5 and 2e-5 respectively) to preserve pre-trained representations, while the subgoal and action heads use higher learning rates (1e-4) to facilitate rapid adaptation to the robotic domain. The training leverages mixed-precision computation with bfloat16 to accelerate training while maintaining numerical stability, and employs gradient accumulation to effectively increase batch size despite memory constraints.

## D    IMPLEMENTATION DETAILS

### D.1    FEATURE POOLING METHODS

The vision–language backbone produces hidden states $\mathbf{H}[k, :, :] \in \mathbb{R}^{n \times d}$ at layer $k$, where $n$ is the token length and $d$ the hidden dimension. We investigate pooling strategies that compress $\mathbf{H}[k, :, :]$ into a compact task-aware feature $f_t^c \in \mathbb{R}^d$. We consider three baselines:

| Hyperparameter | Value |
|---|---|
| # GPUs | 8 |
| Batch size | 32 / GPU |
| Optimizer | AdamW |
| Learning rate schedule | Cosine |
| Vision projector learning rate | $1 \times 10^{-5}$ |
| LLM learning rate | $2 \times 10^{-5}$ |
| Subgoal head learning rate | $1 \times 10^{-4}$ |
| Action head learning rate | $1 \times 10^{-4}$ |
| Warmup ratio | 0.05 |
| Weight decay | 0.01 |
| Precision | bfloat16 |
| Dataloader workers | 16 |
| Gradient accumulation steps | 1 |
| Repeated diffusion steps | 8 |
| Stage I training epochs | 4 |
| Stage II training epochs | 2 |

Table 8: Comprehensive training hyperparameters for SubgoalVLA, including component-specific learning rates, optimization settings, and multi-stage training configuration.

- **Mean Pooling.** Computes the average representation $\bar{f}_t^c = \frac{1}{n} \sum_{j=1}^n \mathbf{H}[k, j, :]$ and selects the $k$ tokens most similar to it, with similarity scores $s_i = \mathbf{H}[k, i, :]^\top \bar{f}_t^c$. Tokens closest to the global mean are prioritized.

- **Max Pooling.** Selects the $k$ tokens with the largest L2 norms $\|\mathbf{H}[k, i, :]\|_2$, emphasizing features with strong activations.

- **Attention Pooling.** Learns a query vector $\mathbf{q} \in \mathbb{R}^d$ and scale $\alpha$ to compute relevance scores $a_i = \alpha \cdot \mathbf{H}[k, i, :]^\top \mathbf{q}$, enabling adaptive token selection based on task relevance.

### D.2 BASELINES

We provide a brief overview of the baseline methods considered in our comparison.

- **HULC** learns a hierarchical policy in which a high-level multimodal transformer encodes observation sequences to infer categorical latent plans, while a low-level policy executes actions conditioned on these latent representations in the gripper camera frame. It employs a contrastive vision–language alignment loss and operates in latent space rather than producing explicit subgoals.

- **PAFF** adapts pre-trained policies to new tasks and environments through a self-supervised feedback loop. The policy first collects demonstrations by "playing" with randomly generated instructions, which may contain errors. A fine-tuned CLIP model with spatio–temporal adapters relabels these trajectories by retrieving appropriate instructions, and the policy is then fine-tuned on the relabeled pairs, enabling adaptation without manual annotation.

- **SuSIE** employs a pre-trained image diffusion model to generate intermediate visual subgoals from language commands. Given an initial observation and instruction, it synthesizes future target images representing task milestones. A low-level goal-conditioned policy then navigates between these visual waypoints, effectively decomposing long-horizon tasks into shorter, visually grounded segments.

- **RT-1** is a 35M-parameter transformer model designed for real-time robotic control. It encodes instructions using a Universal Sentence Encoder and visual observations using an ImageNet-pretrained EfficientNet with FiLM conditioning. TokenLearner compresses visual features before feeding them to a decoder-only transformer that outputs discretized actions. It is trained end-to-end on large-scale multi-task demonstrations without leveraging foundation models.

- **VLAS+** extends standard vision–language–action architectures with multimodal input capabilities. For fair comparison, we adopt its text-based configuration rather than the speech-based variant. VLAS+ employs action tokenization to unify processing of vision, language, and action within a single transformer, trained end-to-end on robot demonstrations.

- **RoboFlamingo** adapts the Flamingo vision–language model to robotic control by adding a specialized action head. This dual-system design preserves the semantic reasoning capabilities of the pre-trained model while enabling robot-specific motor control through the dedicated policy head.

- **DeeR** implements a dual-system architecture with dynamic inference optimization. A high-level multimodal reasoning module is paired with a low-level action execution module, linked by an early-exit mechanism that adjusts computational depth based on task complexity. This design improves efficiency while maintaining performance on tasks requiring complex reasoning.

# E  ADDTIONAL RESULTS

## E.1  QUANTATIVE RESULTS

The success rate for the 1st to 5th tasks in a task chain is shown in Table 9. Methods such as PAFF, SuSIE, RT-1 and VLAS+, they doesn't provide results under settings of $D \to D$.

Table 9: Long-horizon task completion success rates ($\uparrow$) on the CALVIN benchmark using the $D \to D$ split. Methods are grouped by paradigm: latent planning, text-based subgoals, image-based subgoals, single-system VLA, and dual-system VLA. LH-$k$ denotes a chain of length $k$. And **Avg. Len.** denotes the average successful chain length.

| Method | Category | Foundation Model | LH-1 | LH-2 | LH-3 | LH-4 | LH-5 | Avg. Len. |
|---|---|---|---|---|---|---|---|---|
| HULC (RA-L'22) | Latent | - | 0.827 | 0.649 | 0.504 | 0.385 | 0.283 | 2.64 |
| RoboFlamingo (ICLR'24) | Dual system | Flamingo-3B | 0.839 | 0.643 | 0.429 | 0.357 | 0.196 | 2.46 |
| DeeR (NeurIPS'24) | | Flamingo-3B | 0.853 | 0.696 | 0.549 | 0.420 | 0.312 | 2.83 |
| **SubgoalVLA** | Dual system | LLaMA-2 7B | **0.865** | **0.729** | **0.604** | **0.469** | **0.344** | **3.01** |

## E.2  ROLLOUT VISUALIZATION

Figure 8 illustrate qualitative rollouts on two distinct long-horizon (LH-5) task chains from the CALVIN benchmark. In both cases, SubgoalVLA successfully executes all five subtasks in sequence, demonstrating the ability to sustain coherent behavior across extended horizons.

## E.3  TRAINING COST

We employ DeepSpeed acceleration framework to optimize training efficiency and memory utilization. Training of SubgoalVLA with the LlaVA 7B backbone is conducted on 8 NVIDIA A100 80GB GPUs, requiring approximately 36 hours to complete the full training regimen of $4+2$ epochs across the CALVIN dataset's ABCD environments. Our implementation supports multi-node distributed training to enable scalable deployment across larger computational clusters.

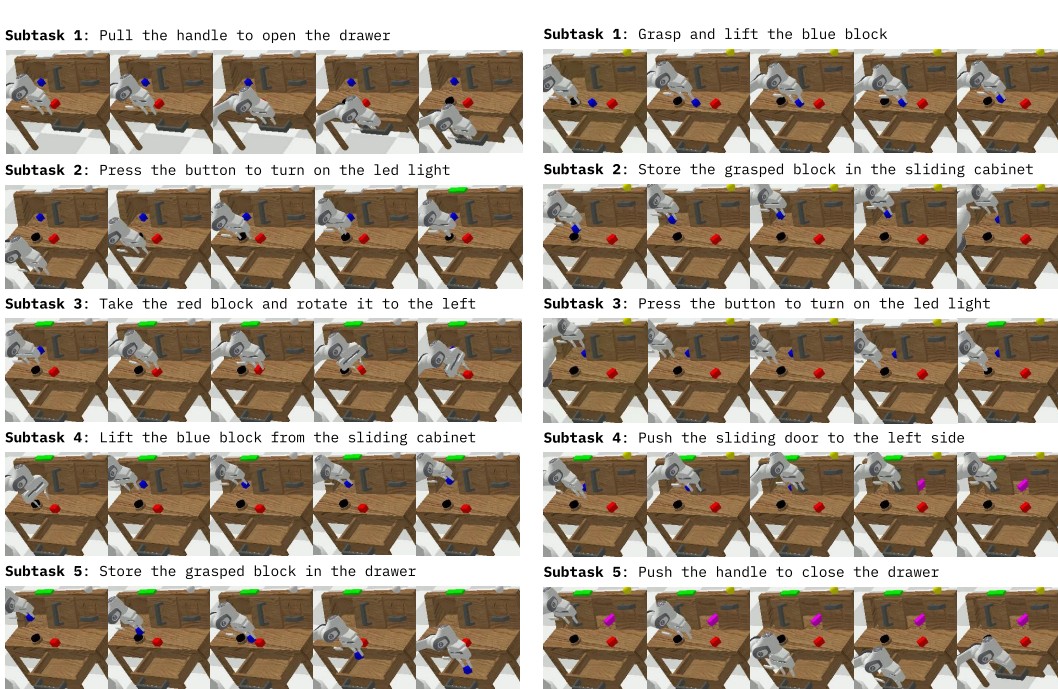

Figure 8: Example rollouts of two LH-5 task chains. (left) open drawer → turn on led → rotate red block left → lift blue block slider → place in drawer. (right) lift blue block table → place in slider → turn on led → move slider left → close drawer.

