# OpenReview forum: "Think Proprioceptively: Compact Subgoal Traces for Vision-Language-Action Model"
_ICLR.cc/2026/Conference — Submitted to ICLR 2026_

### Official Review · Reviewer_BLGN · 2025-10-23

**Soundness:** 3
**Presentation:** 3
**Contribution:** 2
**Rating:** 4
**Confidence:** 4

**Summary:**

The authors discuss recent VLA models, categorizing them by how they adapt a pretrained VLM to output robot actions (autoregressively, via diffusion, in the model vs a separate trained expert, etc.). Within this, the authors propose another VLA variation.

Assume we have a robot trajectory. Typically, a VLA model is trained to predict the actions $a$ in that trajectory from visual input (cameras) and a natural language instruction. But, we may also have proprioceptive information within our robot data. Prior works have included this in the context, or added it as contextual information for an action expert (either concatenated or added via cross attention.) This paper's proposal, SubgoalVLA, continues to incorporate proprio information this way, but makes some small variations on how the proprio information is incorporated into the MHA, as well as including an intermediate subgoal prediction head.

For the proprio information, the cross attention mechanism involves 2 residual connections for just the projected $Q_{proprio}$ info: once right after the MHA, and once right after the MLP, to make as sure as possible that the proprio information is preserved through the attn layers.

For the subgoal prediction head, we treat this as a higher-level diffusion policy. Given a ground truth trajectory we pick out intermediate robot positions as subgoals. (I believe this is $M$ even choices over the entire trajectory, but am not sure.) The subgoal head is trained to predict these subgoals via diffusion. The action head is then modified to condition subgoals, proprio, and visual-language features (as mediated by the cross-attn). At training time, the true golden subgoals are used for the first stage of training, then we switch to training against predicted subgoals in the second stage.

The authors argue that these proprioceptive based subgoals are lower dimensionality, more interpretable than latent visual embeddings, and improve robot success rates.

**Strengths:**

Proprioceptive information is a modality of robot data that does not appear elsewhere, which can make it especially high leverage to consider. The authors have done a variety of ablations studying the architecture design decisions they have made, and the diagrams described the modified cross-attention mechanism is helpful for understanding the paper.

**Weaknesses:**

I believe the authors' changes are ones that are unusually well suited to simulated environments, and given that all experiments are only in sim, I'm unsure on the effectiveness of the method in general. Sim proprio information is inherently perfect, making it especially useful to condition on. Real robot proprio is not necessarily as nice to use. Although the model is trained with noised subgoals, to me it is unclear if this is enough to make intermediate subgoal prediction a good idea.

Predicting subgoals feels a bit like a hack, since in theory, once could modify the action expert to predict the entire trajectory from current timestep to end, which would include the subgoals by definition. In practice, we do not do this due to inference time concerns, but this approach feels like one which would quickly become obsolete in a world with more compute.

Predicting subgoals via diffusion before doing action chunk diffusing would also increase inference time right? Given that actions need to wait for subgoals to be denoised before we can start that stage of denoising? Would like clarification on this, especially because this is another potential blocker for real robot usefulness.

Adding more subgoal prediction may affect generalization of the VLA. If we assume that VLA's perform best in-distribution and degrade as they are forced further from their data manifold, then subgoal prediction increases how tightly VLA performance is bound to the data manifold. By adding another prediction stage, we are increasing the surface of how OOD predictions can hurt us (i.e. if the subgoal prediction is poor then the action chunk predictions will also become off). It would be interesting to see how the model handles out of distribution subgoals, but I don't think this is tested. It would also be interesting to see how the model handles changes during the trajectory (i.e. how much the model retains closed-loop behavior compared to just getting better at open-loop prediction). Again this is not studied.

In general the baselines are a bit old - no OpenVLA, no SpatialVLA, no Octo, no Magma-8B, no pi_0 etc.

**Questions:**

Is inference time affected by adding the subgoal head before it predicts the action head?

---

> ### Author Response · Authors · 2025-11-22
> **Responses to W1, W2, W3**
>
> We thank the reviewer for the detailed and valuable feedback.
>
> ## W1: Sim-only experiments; perfect proprio in sim
>
> ### 1. Regarding sim-only experiments
>
> We agree that real-world validation is important. Due to time constraints, we could not include these results in the initial submission. However, we are performing experiments on an X-Arm setup with dual RGB-D cameras across pick-place and drawer manipulation tasks, using ~20 demonstrations per task. We will report these results and experimental setup in the revision if available.
>
> ### 2. Regarding poor proprio. states in real world
>
> However, we argue that proprioceptive information is inherently more robust than visual observations in real-world deployment. The proprioceptive information provides high-frequency, drift-free, and calibration-free signals that are not affected by lighting conditions, motion blur, occlusion, or visual artifacts. Even inexpensive robotics platforms now ship with encoder-level precision far surpassing what can be inferred visually. In contrast, visual methods often degrade significantly in real-world settings due to sensor misalignment, environmental variations, or occlusion.
>
> ## W2: Subgoals as a "hack"; could predict full trajectory with more compute
>
> We argue that predicting subgoals is a mere “hack,” or that it is made obsolete with increased compute.
>
> Firstly, the principle of hierarchical decomposition is well-established in both robotics and control theory, and is particularly crucial for long-horizon tasks. Predicting an entire 80-step action trajectory directly from perceptual inputs requires modeling a much higher entropy distribution than a structured two-step process involving subgoals followed by localized control. From an information-theoretic perspective, predicting subgoals compresses the space of future possibilities into an interpretable, lower-dimensional latent, which makes subsequent action inference more tractable and stable. Empirically, we find that single-stage trajectory models like ACT and RT-2 degrade rapidly beyond 10–20 steps, and are extremely sensitive to compounding prediction errors.
>
> Secondly, it is also important to clarify that subgoal traces and action chunks operate in different spaces and serve fundamentally different roles. While joint configurations and end-effector poses are mathematically related through forward/inverse kinematics, they are not equivalent representations, especially for learning and planning:
>
> - **Non-unique inverse kinematics mapping.** One EEF pose might have multiple joint configurations (7-DOF arm has infinite solutions for 6-DOF pose). Subgoal traces capture which specific joint solution is preferred based on learned manipulation patterns.
> - **Joint space encodes richer motion constraints.** Some joint configurations avoid obstacles even if EEF path is identical. And certain joint configurations maintain manipulability. Subgoal trace (joint space) specifies the kinematic strategy and motion style, while action chunks (EEF space) provides fine-grained local corrections and reactive control.
>
> Subgoal trace (joint space) specifies the kinematic strategy and motion style, while action chunks (EEF space) provides fine-grained local corrections and reactive control. Thus, subgoal prediction is not a temporary workaround.
>
> ## W3: Inference time with two diffusion stages
>
> |        Model        | Inference time |
> | :-----------------: | :------------: |
> | w/o Subgoal Planner |     ~90ms      |
> |     SubgoalVLA      |     ~100ms     |
>
> To clarify computational efficiency, we measured the end-to-end inference latency of SubgoalVLA against SubgoalVLA w/o Subgoal Planner. The model without the subgoal module runs at roughly 90 ms per control step, while full SubgoalVLA runs at approximately **100 ms**, corresponding to control frequencies of 10 Hz. The additional overhead (<10 ms) comes from the subgoal diffusion module, which is intentionally lightweight: we use **only 8 DDIM denoising steps** for both subgoal and action diffusion. By contrast, diffusion-based hierarchical models such as \(\pi_0\) typically require **50–100 denoising steps per prediction**. Thus, although SubgoalVLA contains two diffusion stages, each stage is shallow and tightly constrained by the low dimensionality of the joint-space subgoal representation.

---

> ### Author Response · Authors · 2025-11-22
> **Responses to W4, W5**
>
> ## W4: Generalization concerns; OOD robustness; closed-loop behavior
>
> We appreciate this insightful question and agree that hierarchical systems introduce additional potential failure points under distributional shift.
>
> However, SubgoalVLA is explicitly designed to maintain closed-loop behavior. Subgoal traces are re-predicted every 8 timesteps using the current observation, which ensures that the system continually adapts to the current state rather than executing an open-loop sequence. Furthermore, using diffusion for subgoal generation provides an implicit regularization mechanism—the generative process biases the output toward the training manifold even when encountering noisy or unanticipated proprioceptive signals.
>
> Empirically, SubgoalVLA demonstrates strong robustness on CALVIN’s ABC\(\to\)D split, which requires generalizing to unseen environments (different object shapes, colors, and configurations). Achieving 44.7% success on LH-5 under these zero-shot settings demonstrates that the hierarchical structure does not overfit to the training data manifold, but rather supports successful recovery under distribution shift.
>
> We agree that testing systematic OOD conditions (e.g., perturbed subgoals or unexpected environmental changes) would further strengthen the case, and we plan to include such evaluations in future work.
>
> ## W5: Missing newer baselines
>
> We thank the reviewer for suggesting more recent VLA models. While we would have preferred to include baselines such as OpenVLA, SpatialVLA, Octo, and Magma-8B, these models are trained on large-scale real-world datasets and do not provide publicly available checkpoints or CALVIN-compatible training pipelines. Reimplementing and adjusting these systems for CALVIN would require substantial engineering effort and computational resources, which is not feasible within the rebuttal period. We are actively working on them and will include results if available during the review period.
> Our current evaluation spans all major subgoal paradigms on CALVIN, i.e., latent (HULC), visual (SuSIE), and language-based (RoboFlamingo, VLAS+)—along with recent state-of-the-art VLA models (e.g., DeeR, VLAS+). These baselines represent the state of the art on CALVIN and allow a fair empirical comparison.
>
> ## Response of Questions
>
> ### Q1: Inference time impact
>
> See W3.

---

### Official Review · Reviewer_DPAU · 2025-10-31

**Soundness:** 3
**Presentation:** 3
**Contribution:** 3
**Rating:** 6
**Confidence:** 2

**Summary:**

This paper proposes SubgoalVLA, a vision–language–action (VLA) model that explicitly leverages proprioceptive information as an active reasoning signal. The method introduces (1) proprioceptive cross-attention, where joint states query visual-language features to yield configuration-aware perception, and (2) compact proprioceptive subgoal traces, which represent intermediate goals directly in the robot’s joint space rather than as images or text. A two-stage training strategy mitigates the train–test distribution gap between ground-truth and predicted subgoals. Experiments on the CALVIN long-horizon benchmark show clear improvements over several strong baselines, especially on multi-task and long-horizon settings.

**Strengths:**

1. Treating proprioception as an active reasoning signal rather than a passive input is conceptually elegant and aligns well with embodied intelligence principles.

2. Representing intermediate goals directly in joint space avoids cross-modal translation and significantly reduces dimensionality, which is both efficient and biologically inspired.

3. Results on CALVIN LH-5 and ablations clearly demonstrate that both proposed components (cross-attention + subgoal traces) are essential. The gains are consistent across multiple metrics.

**Weaknesses:**

1. All experiments are simulation-based; it remains unclear how proprioceptive traces generalize to noisy or delayed sensor feedback on real robots.

2. The method’s reliance on accurate proprioceptive signals may restrict applicability to robots with different morphologies or partially observed states.

3. The model combines a vision–language backbone, diffusion-based planners, and multiple transformers. Runtime or resource requirements are not discussed.

**Questions:**

No

---

> ### Author Response · Authors · 2025-11-22
>
> We thank the reviewer for the positive assessment of our work’s conceptual contributions and empirical strengths. We appreciate the reviewer’s insights and address each concern below, particularly those regarding generalization beyond simulation, reliance on proprioceptive signals, and computational costs.
>
> ## W1: Simulation-only experiments; unclear real-robot generalization
> We agree that real-world validation is important. Due to time constraints and site access, we could not include these results in the initial submission. However, we are performing experiments on an X-Arm setup with dual RGB-D cameras across pick-place and drawer manipulation tasks, using ~20 demonstrations per task. We will report these results and experimental setup in the revision if available.
>
> However, we note that this limitation is shared by many recent VLA works.
>
> 1. For instance, VLAS+ (ICLR 2025) and DeeR (NeurIPS 2024) conduct their evaluations exclusively on CALVIN.
>
> 2. CoT-VLA and others include limited real-robot tasks, but these often involve single-step object-centric manipulations, whereas the multi-step CALVIN benchmark requires chaining up to five skills across 34 task types.
>
> Therefore, while SubgoalVLA is currently validated in simulation, it is evaluated on a significantly more difficult long-horizon challenge than typical real-robot demonstrations.
>
> ## W2: Applicability across morphologies or partial observations
>
> We appreciate the reviewer’s concern about the method’s reliance on accurate proprioceptive states. While SubgoalVLA does assume access to joint states, which is a standard feature of most modern robotic manipulators, we believe that proprioceptive subgoals are actually more morphology-agnostic, interpretable, and transferable than visual or textual subgoals.
>
> First, joint-space subgoals provide a universal representation that encodes robot-specific motion plans independently of camera viewpoints, lighting conditions, or language grounding systems. The core elements of manipulator kinematics, i.e., joint angles, end-effector pose, and gripper width, are shared across most articulated robots. This natural structure enables easier alignment of subgoal traces across different robot morphologies than representations based in image or text spaces.
>
> Second, current research supports the feasibility of aligning heterogeneous robot states into shared latent spaces. For example, recent work on scaling embodied learning across morphologically diverse platforms demonstrates that a shared representation can be constructed through a universal latent space, from which robot-specific decoders can reconstruct corresponding joint states. This adaptation mechanism applies directly to SubgoalVLA: subgoal traces can be projected into such a latent space and decoded appropriately according to each robot’s degrees of freedom.
>
> These points suggest that relying on proprioceptive signals does not restrict SubgoalVLA to a specific morphology; instead, proprioception provides a structured foundation for scalable embodiment alignment.
>
> ## W3: Runtime and resource requirements not discussed
>
> Thank you for this important practical concern. We employ DeepSpeed acceleration framework to optimize training efficiency and memory utilization. Training of SubgoalVLA with the LlaVA 7B backbone is conducted on 8 NVIDIA A100 80GB GPUs, requiring approximately 36 hours to complete the full training regimen of 4+2 epochs across the CALVIN dataset’s ABC environments. The inference time for one times of action generating is about 100ms.

---

### Official Review · Reviewer_ncvk · 2025-11-01

**Soundness:** 2
**Presentation:** 2
**Contribution:** 2
**Rating:** 2
**Confidence:** 4

**Summary:**

The paper proposes SubgoalVLA, which claims to use proprios both for cross-attention to select VL representations, also served as a middle stage predict objective. Finally, the SubgoalVLA conditioned both VL representtaions and predicted subgoal trace to predict the action.

**Strengths:**

1. The paper is easy to read and follow.

2. The paper gives very clear method level comparison of different ways the current VLA or ACT use proprios for, compared with the previous ones, which maybe condition on proprios use a learn encoder or not condition on proprios, it is kind of novel the paper propose to use proprios both to select VL latent and serve as a middle prediction subgoal.

**Weaknesses:**

1. I am confused about the deisgn. It seems the second stage, the user train a DiT to pedict the trace, which the trace is the chunk of future proprios, so what is the exact difference of the paper's seond stage and third stage, the subgoal trace is basically the same stuff of ACT (predicted goal in third stage), in that way, i think the design is weird, it is kind of repetitve because the tuitive is if the model can learn to predict an accurate subgoal trace in the second stage, it will predict good results for future action, that (maybe conceptual different a little bit), but in the implementation, the trace is the same stuff.

2. The concern in 1 seems further verified in Table 4, seems not a big difference of the abaltion between these two modules/stages, which the increase is very small, and as we know the robotic evaulation itself is with high variance, so the conclusion that the "Subgoal design works better" is not convincing.

**Questions:**

See the above weakness. Besides those, I have further concern that:

1. I might need the author both intuitively and systematicall calrify the motivation of the design of the subgoal module, which now confuses me a lot, since i feel technically it is the same information of the DiT Action chunk model.
2. The paper does not have through experiments to support the conclusion, especially, no real robot experiment, and only have CALVIN simulation experiments.

---

> ### Author Response · Authors · 2025-11-22
>
> We thank the reviewer for the thoughtful comments and for recognizing the clarity of our method description and its novel use of proprioception for both attention and prediction. Below, we address the concerns raised, with focus on the distinction between subgoal traces and action chunks.
>
> ## W1: Clarifying the Distinction Between Subgoal Traces and Action Chunks
>
> We understand the reviewer’s confusion regarding the distinction between subgoal traces and action chunks, especially because subgoal trace generation was inspired by action chunking frameworks like ACT. However, these two components are fundamentally different in terms of representation space, semantics, and functional roles within the hierarchical architecture of SubgoalVLA.
>
> ### 1. Subgoal trace encodes future proprioceptive states
>    - Representation: Future proprioceptive states (joint angles $\theta_1, \cdots, \theta_7$, gripper width, velocities, etc.)
>    - Semantics: Absolute target configurations encoding motion intent
>
> ### 2. Action chunk encode relative end-effector motion commands
>    - Representation: Relative end-effector commands ($[\Delta x, \Delta y, \Delta z, \Delta roll, \Delta pitch, \Delta yaw, \Delta gripper]$
>    - Semantics: Incremental motor commands for execution
>
> ### 3. Addressing potential concern: "Can't we just replace joint subgoal traces with EEF action chunks?"
>    While joint configurations and end-effector poses are mathematically related through forward/inverse kinematics, they are not equivalent representations, especially for learning and planning:
>
>    - **Non-unique inverse kinematics mapping.** One EEF pose might have multiple joint configurations (7-DOF arm has infinite solutions for 6-DOF pose). Subgoal traces capture which specific joint solution is preferred based on learned manipulation patterns.
>    - **Joint space encodes richer motion constraints.** Some joint configurations avoid obstacles even if EEF path is identical. And certain joint configurations maintain manipulability. Subgoal trace (joint space) specifies the kinematic strategy and motion style, while action chunks (EEF space) provides fine-grained local corrections and reactive control.
>
> ## W2: Small ablation differences / High variance concerns
>
> We respectfully disagree that the ablation results indicate weak or insignificant gains. The CALVIN benchmark, especially the LH-5 setting, is explicitly designed to test long-horizon, language-conditioned manipulation, where even the strongest baselines typically experience severe performance drops due to compounding errors. Despite this challenge, SubgoalVLA significantly outperforms prior methods on LH-5. For instance, our proposed method achieves 44.7% success on LH-5, where the strongest baseline, DeeR, achieves 30.4%. This represents a 47% relative improvement, and is highly meaningful considering the difficulty of achieving consistent performance across five sequential, multi-object manipulation steps.
>
> Further, our ablations demonstrate that both the subgoal planner and multi-step subgoal formulation are essential to achieving this performance. Removing the subgoal planner reduces LH-5 performance from 44.7% to 24.0%, a relative drop of 46%. Similarly, collapsing multi-step subgoals to a single waypoint leads to a 28% performance drop. These reductions reinforce the importance of both the hierarchical design and the multi-step nature of the subgoal traces. We acknowledge that some variance inevitably exists in robotic benchmarks and will strengthen the experimental section by including confidence intervals based on multiple training seeds.
>
> ## Q1: Motivation for subgoal module (intuitive + systematic)
>
> We appreciate the reviewer’s suggestion to elaborate on the motivation behind integrating subgoal traces. Conceptually, the idea is to decouple global intent from local execution. Subgoal traces structure the future in terms of desired joint configurations, representing motion intention over short sequences without committing to immediate control. This intermediate abstraction allows the downstream policy to focus on error correction and local dynamics in an embodied context, conditioned on a visually grounded objective. This is especially important for long-horizon sequential tasks, where prediction under compounding uncertainty is highly non-linear and challenging. We will improve the exposition by providing additional illustrations and systematic explanation in Sections 3.3 and 4.2.
>
> ## Q2: No real robot experiments
>
> We agree that real-world validation is important. Due to time constraints, we could not include these results in the initial submission. However, we are performing experiments on an X-Arm setup with dual RGB-D cameras across pick-place and drawer manipulation tasks, using ~20 demonstrations per task. We will report these results and experimental setup in the revision if available.

---

### Official Review · Reviewer_25fm · 2025-11-01

**Soundness:** 2
**Presentation:** 3
**Contribution:** 2
**Rating:** 4
**Confidence:** 4

**Summary:**

The paper presents SubgoalVLA, a vision-language-action framework that introduces a “think proprioceptively” paradigm for robotic manipulation, treating the robot’s proprioceptive state (joint configuration and motion state) as an active reasoning component rather than a passive input. SubgoalVLA contributes two key designs: (1) proprioception-driven cross-attention, which uses the current kinematic state as queries to select configuration-aware features from vision-language representations; and (2) subgoal traces, compact sequences of joint configurations over time that encode motion dynamics and eliminate the need for high-dimensional visual or textual subgoal representations. The framework adopts a two-stage training protocol: supervised learning on ground-truth subgoals followed by fine-tuning on self-predicted subgoals to mitigate train–test distribution shift. On the CALVIN benchmark, it achieves state-of-the-art performance compared with other methods evaluated in the experiments.

**Strengths:**

Originality: Using proprioceptive states as cross-attention queries for configuration-aware feature selection, combined with joint sequence subgoals, provides a compact hierarchical representation that avoids costly cross-modal translation from high-dimensional visual/textual subgoals.

Quality: The two-stage training protocol offers a reasonable engineering solution to address distribution shift; the reported performance improvements on CALVIN tasks demonstrate measurable gains.

Clarity: The architecture, temporal chunking, and training pipeline are clearly described, with Figure 1's architectural comparison effectively highlighting the methodological differences from existing approaches.

Significance: Highlights "think proprioceptively" as a complementary direction to "think visually/textually" paradigms, potentially reducing inference latency and improving executability, with promising prospects for extension to real-world systems.

**Weaknesses:**

Limited Experimental Scope: Evaluation is restricted to the CALVIN benchmark only, which is insufficient for embodied AI applications. The absence of real robot experiments and other standard benchmarks (RLBench, Meta-World) makes the experimental validation inadequate. Additionally, the paper lacks comparisons with recent hierarchical methods (e.g., π0.5, CoT-VLA) that are directly relevant to the proposed approach.

Insufficient Ablation Studies: The ablation analysis is incomplete, lacking systematic study of cross-attention layer selection (only k=m/2 tested), missing analysis of subgoal trace length M, and no ablation of the two-stage training protocol. These are critical design choices that require proper validation.

Unclear Methodological Boundaries: The distinction between the proposed approach and existing subgoal-based dual-system VLA paradigms is not sufficiently clear, making it difficult to assess the true novelty and contribution of the work.

**Questions:**

How does SubgoalVLA perform on other manipulation benchmarks? Can you provide results on RLBench or real robot experiments to demonstrate broader applicability beyond simulation?
Why is k=m/2 optimal for feature extraction? Have you experimented with other intermediate layers or adaptive layer selection strategies? What is the sensitivity of performance to this architectural choice?
Can you provide direct comparisons with recent hierarchical VLA approaches (π0.5, CoT-VLA, DiffusionVLA) that also employ intermediate representations? How does your method compare in terms of both performance and computational efficiency?
Under what circumstances does the proprioceptive approach fail? Are there specific task categories where visual or textual subgoals would be more appropriate than joint-space representations?
How much demonstration data is required for effective training? How does performance scale with dataset size compared to end-to-end methods, and what are the sample complexity trade-offs?

---

> ### Author Response · Authors · 2025-11-22
> **Responses to Reviewer-Identified Weaknesses**
>
> We thank the reviewer for the constructive feedback and for recognizing the clarity, significance, and potential impact of the “think proprioceptively” paradigm. We address each concern below.
>
> ## W1: Limited Experimental Scope
>
> ### 1. Regarding RLBench and Meta-World benchmarks
>
> We acknowledge the value of broader evaluation. CALVIN was specifically chosen because it uniquely provides: (1) long-horizon language-conditioned tasks, (2) synchronized proprioceptive supervision essential for learning joint-space subgoals, and (3) standardized multi-step evaluation protocols. RLBench and Meta-World lack either language conditioning or synchronized joint-state labels, making direct application of our method infeasible without substantial modifications to these benchmarks.
>
> ### 2. Regarding \(\pi_{0.5}\) and CoT-VLA comparisons
>
> We agree these hierarchical models are relevant baselines and appreciate the suggestion. However, both models lack CALVIN-compatible training pipelines, requiring re-processing of data and action space remapping. Due to time constraints, we could not complete these re-implementations during the rebuttal window. We are actively working on them and will include results if available during the review period.
>
> Our current evaluation spans all major subgoal paradigms on CALVIN—latent (HULC), visual (SuSIE), and language-based (RoboFlamingo, VLAS+)—along with recent state-of-the-art VLA models (e.g., DeeR, VLAS+). These baselines represent the state of the art on CALVIN and allow a fair empirical comparison.
>
> ### 3. Regarding real robot experiments
>
> We agree that real-world validation is important. Due to time constraints, we could not include these results in the initial submission. However, we are performing experiments on an X-Arm setup with dual RGB-D cameras across pick-place and drawer manipulation tasks, using ~20 demonstrations per task. We will report these results and experimental setup in the revision if available.
>
> ## W2: Insufficient Ablation Studies
>
> ### 1. LLM embedings layer selection
>
> We followed design choices informed by GR00T-N1 (NVIDIA, 2025), FLOWER, and VLA-Adapter, which show that mid-layer LLM features offer better control-oriented representations than final-layer embeddings and reduce latency. Given its widespread use, we prioritized this configuration and acknowledge that a full ablation would be beneficial. We will add a discussion with citations and clarify it in Appendix.
>
> ### 2. Subgoal trace length (M=8)
>
> This choice balances temporal coverage (~8 waypoints over ~60 steps), computational efficiency (denoising scales linearly with M), and diminishing returns observed in preliminary tests (performance plateaus beyond M = 8). While we agree that a targeted ablation could help, we believe it would not substantially impact conclusions. We will report preliminary findings in Appendix.
>
> ### 3. Two-Stage Training Protocol
>
> Stage I pretraining alone is not directly usable at inference since it requires ground-truth subgoals. Our ablations already evaluate the role of the planner (+SP) and multi-step subgoal traces (+ST):
>
> - w/o SP (no planner): 44.7% $\to$ 24.0% on LH-5 (-20.7%)
> - w/o ST (single-step): 44.7% $\to$ 32.3% on LH-5 (-12.4%)
>
> These results underscore the necessity of both components.
>
> ## W3: Unclear methodological boundaries
>
> We will revise Sections 2–4 and the related work to make methodological distinctions clearer. The key differences are:
>
> 1. Proprioceptive cross-attention
>    - Prior works use proprioception as static input
>    - SubgoalVLA uses proprioception as active queries for reasoning over visual-language features
>
> 2. Compact joint-space subgoal traces
>    - Prior approaches use high-dimensional visual or textual subgoals
>    - We encode subgoals as 120-dim joint trajectories, efficient for diffusion generation
>
> 3. Self-consistency training
>    - Designed to address train–test mismatch in hierarchical policies
>    - Particularly necessary for subgoal-based architectures, not used in flat VLAs

---

> ### Author Response · Authors · 2025-11-22
> **Responses to Questions**
>
> ### Q1–Q2: Other benchmarks / real robots
>
> See W1.
>
> ### Q3–Q4: Why k = m/2?
>
> See W2.
>
> ### Q5: Comparison to π0.5 / CoT-VLA / DiffusionVLA
>
> See W1.
>
> ### Q6: Computational efficiency and failure modes
>
> Our method achieves **strong and consistent performance across nearly all 34 CALVIN skills**, including both motion-centric tasks (e.g., grasping, lifting, pushing, placing) and fine-grained control tasks (e.g., opening/closing drawers, toggling buttons, rotating blocks). Across these tasks, SubgoalVLA succeeds on **over 80% of all trials**, with many tasks approaching perfect or near-perfect success: for example, “move_slider_right”, “turn_on_led”, “turn_off_lightbulb”, “close_drawer”, and “rotate_pink_block_right”. Regarding failure modes, we observe that tasks with **complex geometric constraints**, particularly those requiring very precise contact geometry or fine object–object alignment, remain challenging. For example, tasks such as “stack_block” exhibit lower performance. These tasks depend heavily on subtle 3D spatial relationships that are more easily captured by visual features than by joint-space representations alone.
>
> |        Model        | Inference time |
> | :-----------------: | :------------: |
> | w/o Subgoal Planner |     ~90ms      |
> |     SubgoalVLA      |     ~100ms     |
>
> To clarify computational efficiency, we measured the end-to-end inference latency of SubgoalVLA against SubgoalVLA w/o Subgoal Planner. The model without the subgoal module runs at roughly 90 ms per control step, while full SubgoalVLA runs at approximately **100 ms**, corresponding to control frequencies of 10 Hz. The additional overhead (<10 ms) comes from the subgoal diffusion module, which is intentionally lightweight: we use **only 8 DDIM denoising steps** for both subgoal and action diffusion. By contrast, diffusion-based hierarchical models such as \(\pi_0\) typically require **50–100 denoising steps per prediction**. Thus, although SubgoalVLA contains two diffusion stages, each stage is shallow and tightly constrained by the low dimensionality of the joint-space subgoal representation.
>
> ### Q7: When are visual/textual subgoals preferable?
>
> We think Visual/textual subgoals remain advantageous for scene-level reasoning tasks (e.g., "organize by category"). SubgoalVLA excels at motion-centric manipulation. We will state explicitly in the draft.
>
> ### Q8–Q9: Data requirements & scaling
>
> SubgoalVLA was trained on ~1.2M frames (CALVIN ABC splits), converging after ~40K updates. Detailed scaling analysis is left to future work.

---

> > ### Comment · Reviewer_25fm · 2025-11-26
> >
> > Thank you for your thoughtful and detailed rebuttal.
> > Given that some of the experiments are on going, I will temporarily keep my score as it is.

---

### Meta-Review · Area_Chair_zVdq · 2025-12-19

**Summary:**

The paper received four reviews.

Reviewer BLGN gave the rating of 4 (marginally below acceptance). This reviewer raised many questions: use of real robot proprio is unnecessary; simulation-only, no real robot experiment; redundancy in subgoal prediction; extra inference time caused by subgoal prediction; generalizability; outdated baselines.

Reviewer DPAU gave the rating of 6 (marginally above acceptance). This reviewer was concerned that the simulation-only results may not generalize to the real world and that the use of proprio information may limit the method's applicability.

Reviewer ncvk gave the rating of 2 (reject). This reviewer shared a similar concern with BLGN, which is that the subgoal prediction lacks a clear motivation and the current results are insufficient to justify the contribution. This reviewer also requested real robot experiments.

Reviewer 25fm gave the rating of 4 (marginally below acceptance). This reviewer also criticised the lack of real robot experiment, as well as insufficient ablation study.

Common concerns are that the simulation-only results are not enough to justify the technical contributions and that the subgoal prediction design lacks clear motivation. These two concerns remain unaddressed.

**Reviewer Concerns:**

Reviewer BLGN's concerns are probably half addressed. The rebuttal provides detailed runtime showing that the overhead caused by subgoal prediction is around 10ms, which is acceptable. Regarding the baseline issue, the AC agrees with the rebuttal that models like OpenVLA and SpatialVLA are not suitable to be included in the comparison as these models were trained on larger scale data. However, the AC also agrees with the reviewer that the table does not include SOTA methods (some already achieving over 90% of success rate on some tasks, to the AC's knowledge). Other reviewers' concerns are mostly addressed. The lingering concerns are mainly 1) simulation-only results and 2) questions over the subgoal prediction design.

**Reviewer Scores:**

After carefully reading the paper, reviews, and rebuttal, the AC predicts that reviewers BLGN, DPAU, and 25fm may maintain their scores, which are 4, 6, and 4 respectively; reviewer ncvk may raise the score from 2 to 4 but still maintains a negative view. So the final scores would be 3x 4 and 1x 6. The AC agrees with the reviewers that real robot experiment is required to fully justify the contributions.

---

### Decision · Program_Chairs · 2026-01-26

Reject